# Pharmaceutical company payments to dermatology Clinical Practice Guideline authors in Japan

Anju Murayama[1,2]*, Akihiko Ozaki[2], Hiroaki Saito[3], Toyoaki Sawano[4], Yuki Shimada[5], Kana Yamamoto[6], Yosuke Suzuki[2], Tetsuya Tanimoto[2]

1 Tohoku University School of Medicine, Sendai, Miyagi, Japan, 2 Medical Governance Research Institute, Minato-ku, Tokyo, Japan, 3 Department of Gastroenterology, Sendai Kousei Hospital, Sendai, Miyagi, Japan, 4 Department of Surgery, Sendai City Medical Center, Sendai, Miyagi, Japan, 5 Department of Neurosurgery, Minamisoma Municipal General Hospital, Minamisoma, Fukushima, Japan, 6 Department of Internal Medicine, Navitas Clinic, Tachikawa, Tokyo, Japan

* ange21tera@gmail.com

**Data Availability Statement:** Payment data of each CPG authors were available from each pharmaceutical company webpages (http://www.jpma.or.jp/tomeisei/guideline/). And dermatology

## Abstract

Clinical Practice Guidelines (CPGs) play significant roles in most medical fields. However, little is known about the extent of financial Conflicts of Interest (FCOIs) related to pharmaceutical companies (Pharma) selling dermatology prescription products and dermatology CPG authors in Japan. The aims of this study were to elucidate the characteristics and distribution of payments from Pharma to dermatology CPG authors in Japan, and to evaluate the extent of transparency and accuracy in their FCOI disclosures. We analyzed the records of 296 authors from 32 dermatology CPGs published by the Japanese Dermatological Association from the beginning of 2015 to the end of 2018. Using the payment data reported by 79 Pharma between 2016–2017 in Japan, we investigated the characteristics of the CPG authors and the payments from the Pharma to them. Furthermore, we evaluated the transparency and accuracy of the FCOI disclosures of the individual CPG authors. Of the 296 CPGs authors, 269 authors (90.6%) received at least one payment from the Pharma. The total monetary value of payments for the 2-year period was $7,128,762. The median and mean monetary value of payments from the Pharma reporting were $10,281 (interquartile range $2,796 -$34,962) and $26,600 (standard deviation $40,950) for the two years combined. Of the 26 CPG authors who disclosed FCOIs due to the monies received from Pharma, only the atopic dermatitis CPG authors and the acne vulgaris CPG authors published their potential FCOIs. In Japan, most dermatology CPG authors received financial payments from Pharma. The transparency of the CPGs, as reported by the CPG authors, was inadequate, and a more rigorous framework of reporting and monitoring FCOI disclosure is required to improve the accuracy and transparency with relation to possible Conflicts of Interest.

CPGs which we used in this study were publicly available from the Japanese Dermatological Assocciation webpage (https://www.dermatol.or.jp/modules/guideline/index.php?content_id=2). All relevant data are within the manuscript and its Supporting Information files.

**Funding:** This study was supported from the Medical Governance Research Institute, Ain Pharmaciez and the Waseda Chronicle. The funders had no role in study design, data collection and analysis, decision to publish, or preparation of the manuscript.

**Competing interests:** Our institute, Medical Governance Research Institute, received donation from Ain Pharmaciez. Also Dr. Saito received personal fees from TAIHO Pharmaceutical Co., Ltd. outside the scope of the submitted work. Drs. Ozaki and Tanimoto received personal fees from Medical Network Systems outside the scope of the submitted work. This donation from Ain Pharmaceiz does not alter our adherence to PLoS ONE policies on sharing data and materials.

## Introduction

There is increasing global attention on transparency with respect to the financial relationships between the pharmaceutical companies (Pharma) and healthcare professionals, with increasing concern about corrupt or unethical behavior. Although the primary objective of physicians is to respect the best interest of patients, financial relationships with the Pharma can bias a physician's decision regarding the treatment and management of their patient, including drug selection [1,2]. Moreover, financial relationships with the Pharma may cause other forms of corruption, including scientific misconduct, as accentuated in the case of the Valsartan Scandal in Japan [3]. In Japan, the Japan Pharmaceutical Manufacturers Association (JPMA) developed a transparency guideline in the year 2011, with all member companies from 2013 onwards required to voluntarily publish all payments made to physicians, including for lecturing, writing and consultancy work, itemizing the value of payments along with individuals' names and affiliations [4].

Among various medical fields in Japan, dermatology attracts one of the largest amounts of payment from the Pharma. Indeed, our previous research elucidated that Executive Board members of the Japanese Dermatological Association (JDA) received the second highest payments in the median values among those representing 18 basic medical fields in Japan [5]. The plausible explanations of this is a large market size (JPY 203 billion (US$186 million) in 2016) and development of novel and expensive biologic therapies over the last decade, such as ustekinumab (STELARA, approval year 2011), adalimumab (HUMIRA, approval year 2016), secukinumab (COSENTYX, approval year 2014) and brodalumab (LUMICEF, approval year 2016), as well as novel ointments.

Clinical Practice Guidelines (CPGs) stipulate official statements and recommendations concerning clinical questions and treatment options relating to specific diseases [6]. Thus, the authors of CPGs often become attractive targets for Pharma with commercial interests in the specific diseases, both in Japan and globally [7–9] as a possible means of influencing the contents of CPGs to the eventual financial benefit of their own companies [10,11]. Pharma making payments may be aggressively and unethically promoting the sale and use of their drugs. Therefore, we hypothesized that, by examining dermatology CPG authors in Japan who received financial payments from Pharma, we could elucidate whether the amount of money received would be higher among those authoring a larger number of CPGs or in cases where the CPGs recommended the use of drugs newly marketed by the companies making the payments.

## Methods

### Study setting

The JDA was established in 1900 and had 12,080 general members as of March 2019. It is regarded as the primary professional medical society in the Japanese clinical dermatology field with the following roles: to publish academic journals in both English and Japanese, to operate Certification Board examinations in dermatology, and to publish official dermatology CPGs. On the website of the JDA, we obtained 45 dermatology CPGs, complete with their identified authors, that were freely and publicly available as of December 2019. We considered the 32 guidelines published from the beginning of 2015 to the end of 2018 (S1 Table), following a previous study [12].

### Data collection

We collected data on the CPG authors' names, gender, medical specialties, affiliations, and positions at their affiliations, using the CPGs', institutional and other websites. Data of their

FCOI disclosures were obtained as published in the CPGs, and we categorized them in the following three groups; no disclosure, disclosure with aggregated data, and disclosure with individual details. We extracted individual details of FCOI disclosure when available to evaluate its consistency with the database specifying the companies that reported payments to individuals and the amount of payments. Data collection related to payment from pharmaceutical companies are described in the following section.

## Payment source

Payment data were published on the website of each company which was, at the time, a member of the Japan Pharmaceutical Manufacturing Association (JPMA). We collected the payment data from the 78 and 73 companies (79 companies in total) which belonged to the JPMA in 2016 and in 2017, respectively, as in our previous study [13].

Using the collected data, we generated a unified single database, as follows. First, because no data were published in the form of a spreadsheet or in any standardized fashion, data with character codes were converted into a spreadsheet format. Second, data with no character code were converted into text files using an optical character reader (Yomitori kakumei, version 15; Panasonic Solution Technologies Company, Ltd, Tokyo, Japan). Third, for data protected against any form of reproduction, we used FullShot, version 10 software (Inbit Inc, California, USA) to scan the data and convert the resultant images into text files. Finally, we confirmed that the transformed data were accurately converted by comparing them with the original data. Our database included the names of all individual physicians, their primary affiliated institutions, the amounts of payments made by Pharma, and the forms for the payments. The form of payments used was limited, being categorized into the following three types: payment for lectures, payment for authoring, editing, etc., and consulting fees. The data did not include research payments, meal and the benefits, because the Pharma concerned did not report these as separate, identifiable payments [3].

From the payment database, we extracted payment data reported by each company as having been paid to each individual physician by matching individual names using the Excel function "iferror" and "vlookup". For each person named in the database we checked to find and remove any and all duplicates. For each person named in the database we checked to find and remove any and all duplicates. For each name included, we also identified the work affiliation specified by the company making a payment and the area and/or specialty of the individuals concerned. We also visited the websites of their main places of work and, where possible, found biographies and photos of the individuals concerned to confirm the identity of the CPG authors.

We used data on physicians' names, their main work affiliations, the amount of payments, payment formats, and the total number of payments from our payment database. The form of payments was categorized into 3 types: lecturing, writing work, and consulting fees.

## Data analysis

We calculated the proportion of authors who received at least one payment and the mean and median value of payments among all authors of each CPG. In the calculation of mean and median payments, we included the zero values. Furthermore, we analyzed the distribution of payments from Pharma to CPG authors using the Gini index (GI). The GI measures inequality of income or distribution among a given population. The GI ranges from 0 to1, and the greater the GI is, the greater the disparity in the distribution of payments.

Further, using a multivariate negative binomial regression model, we subsequently examined potential factors associated with the monetary value of the payments to CPG authors,

including their gender, work affiliation and the number of involved CPGs. We divided institutional places of work into three categories: universities or university hospitals (professors); universities or university hospitals (non-professors); and other type of institutions. We classified appointed professors and emeritus professors alongside ordinary professors. Designations of 'universities' or 'university hospitals' included CPG authors who worked in a university or a university hospital, and 'others' compromised institutions including CPG authors working in a clinic, research institute or non-university type of hospital. For this analysis, we excluded eight non-physician CPG authors. All payment data was rounded down as a unit of 1 million Japanese yen (US$9,191).

To elucidate the extent of FCOI disclosures, we descriptively analyzed the FCOI policies in the CPGs. When possible, we assessed the accuracy of the FCOI disclosure among the authors, on an individual basis, by comparing their disclosure in the 2016 CPG against our 2016 and 2017 payment database.

Japanese yen were converted to American dollars using the 2016 and 2017 average monthly exchange rate of 108.8 yen and 112.1 yen per US$1 respectively. All analyses were conducted using Microsoft Excel, version 16.0 (Microsoft Corp) and Stata version 15 (StataCorp).

### Ethical clearance

This study was approved by the Institutional Review Board of the Medical Governance Research Institute. Informed consent from the CPG authors was exempted because all the data in this study were publicly available. We followed the Strengthening the Reporting of Observational Studies in Epidemiology (STROBE) reporting guideline.

### Results

We considered all the 296 CPG authors, and Table 1 summarizes their characteristics. Among them, 247 were men (83.4%), 99 (33.5%) were university professors, and 231 (78.0%) were dermatologists. With respect to the CPGs, 61 authors (20.6%) worked on more than one guideline.

Table 2 showed characteristics of pharmaceutical payments made to dermatology CPG authors. There were 7,562 total payments and the total amount paid was $7,128,762, including $5,647,002 (79.2%) for speaking, $484,213 (6.8%) for writing work and $922,495 (12.9%) for consultancy between 2016 and 2017. The median payment value was $10,281 (interquartile range [IQR], $2,796–34,962), and the mean (SD) payment amount was $26,600 ($40,950) per author. The mean value for men was $26,562 (SD, $42,194), compared with $11,278 (SD, $17,529) for women. Of all 296 CPG authors, 269 authors (90.6%) received at least one payment as identified by Pharma reports; $1,000 or more in the case of 241 authors (81.1%) and $100,000 or more for 13 authors (4.4%) for the combined total of 2016 and 2017. Of 79 Pharma that disclosed individual payments, 68 (86.1%) reported making at least one payment to the CPG authors between 2016 and 2017. The amount paid by each Pharma was largest for Maruho Co Ltd ($1,361,417), followed by Mitsubishi Tanabe Pharma Corporation ($657,084), and Taiho Pharmaceutical Co. Ltd ($478,936).

Fig 1 shows the median values, highest payments and percentage of authors who received payments for each CPG. All the CPG authors received at least one payment in 13 CPGs (40.6%). The median value of the payments was largest for authors of the CPG for hand eczema ($72,143). Details of the payments for each CPG are listed in S2 Table.

The GI for total payments was 0.69, suggesting a significant inequity in the distribution of payments among the authors. With respect to amounts reportedly paid, the top 10% and top

**Table 1. Characteristics of dermatology Clinical Practice Guideline (CPG) authors.**

| Variable | Authors (n = 296) No. (%) |
| --- | --- |
| **Affiliations** | |
| Universities | 193 (65.2) |
| University hospitals | 19 (6.4) |
| Other types of hospitals | 61 (20.6) |
| Research institutes | 4 (1.4) |
| Clinics | 19 (6.4) |
| **University professors** | |
| Yes | 99 (33.5) |
| No | 197 (66.5) |
| **Sex** | |
| Male | 247 (83.4) |
| Female | 49 (16.6) |
| **Specialty** | |
| Dermatology | 231 (78.0) |
| Rheumatology and Clinical Immunology | 13 (4.4) |
| Ophthalmology | 10 (3.4) |
| Neurology | 8 (2.7) |
| Pediatrics | 5 (1.7) |
| Cardiovascular internal medicine | 3 (1.0) |
| Orthopedics | 3 (1.0) |
| Pathology | 3 (1.0) |
| Radiology | 2 (0.7) |
| Plastic surgery | 1 (0.3) |
| Chest surgery | 1 (0.3) |
| Respiratory medicine | 1 (0.3) |
| Otolaryngology | 1 (0.3) |
| Gastroenterology | 1 (0.3) |
| Oncology | 1 (0.3) |
| Urology | 1 (0.3) |
| Neurosurgery | 1 (0.3) |
| Cardiovascular surgery | 1 (0.3) |
| Public health | 1 (0.3) |
| Non-physician | 8 (2.7) |
| **Number of CPG worked on** | |
| 10 | 1 (0.3) |
| 6 | 2 (0.7) |
| 5 | 3 (1.0) |
| 4 | 9 (3.0) |
| 3 | 9 (3.0) |
| 2 | 37 (12.5) |
| 1 | 235 (79.4) |

50% of CPG authors received $3,529,929 (49.5%) and 94.8% ($6,757,572) of the total (S1 Fig). Among the top 10%, 23 (79.3%) were university professors.

Table 3 shows the top Pharma with respect to the reported payments for the top five highest-paid CPGs. All the listed Pharma manufactured and sold drugs for the conditions covered by each specific CPG.

**Table 2. Characteristics of pharmaceutical company payments to authors of dermatology Clinical Practice Guideline.**

| Variables | 2016 | 2017 | Combined total (2016 and 2017) |
|---|---|---|---|
| **Total payments** | | | |
| Japanese yen (¥) | 369,453,178 | 418,475,238 | 788,756,266 |
| American dollars ($) | 3,395,709 | 3,733,053 | 7,128,762 |
| **Median (interquartile range)** | | | |
| Japanese yen (¥) | 449,039 (77,959–1,376,265) | 789,482 (257,466–2,242,730) | 1,136,343 (308,844–3,866,285) |
| American dollars ($) | 4,127 (717–12,649) | 7,043 (2,297–20,007) | 10,281 (2,796–34,962) |
| **Mean (standard deviation)** | | | |
| Japanese yen (¥) | 1,248,153 (2,174,180) | 1,743,647 (2,421,027) | 2,943,120 (4,523,970) |
| American dollars ($) | 11,472 (19,983) | 15,554 (21,597) | 26,600 (40,950) |
| **Authors receiving payment** | | | |
| Any | 248 (83.8) | 240 (80.8) | 269 (90.6) |
| ≥ $1,000 | 214 (72.3) | 211 (71.0) | 241 (81.1) |
| ≥ $10,000 | 86 (29.1) | 97 (32.7) | 137 (46.1) |
| ≥ $50,000 | 15 (5.1) | 16 (5.4) | 45 (15.2) |
| ≥ $100,000 | 3 (1.0) | 3 (1.0) | 13 (4.4) |
| **No. of companies making payment, No. (%)** | 65 (83.3) | 59 (78.7) | 68 (87.2) |
| **Median (interquartile range)** | | | |
| Japanese yen (¥) | 1,931,823 (630,710–5,969,875) | 2,390,003 (800,000–9,457,526) | 3,714,189 (818,570–14,478,594) |
| American dollars ($) | 17,756 (5,797–54,870) | 21,320 (7,136–84,367) | 33,586 (7,451–130,442) |
| **Mean (standard deviation)** | | | |
| Japanese yen (¥) | 5,733,890 (10,827,720) | 7,298,963 (12,315,075) | 11,686,714 (22,092,297) |
| American dollars ($) | 52,701 (99,519) | 65,111 (109,858) | 105,735 (199,929) |
| **Ranking of top five contributing pharmaceutical companies, ¥ ($)** | | | |
| 1 | Maruho Co. Ltd | Maruho Co. Ltd | Maruho Co. Ltd |
| | 74,341,244 (683,283) | 76,018,742 (678,133) | 150,359,986 (1,361,417) |
| 2 | Mitsubishi Tanabe Pharma Corporation | Mitsubishi Tanabe Pharma Corporation | Mitsubishi Tanabe Pharma Corporation |
| | 34,955,666 (321,284) | 37,643,172 (335,800) | 72,598,838 (657,084) |
| 3 | Kyowa Kirin Co. Ltd | Taiho Pharmaceutical Co. Ltd | Taiho Pharmaceutical Co. Ltd |
| | 24,220,698 (222,617) | 36,677,348 (327,184) | 53,187,957 (478,936) |
| 4 | Novartis Pharma K.K. | Kyowa Kirin Co. Ltd | Kyowa Kirin Co. Ltd |
| | 20,592,333 (189,268) | 27,262,141 (243,195) | 51,482,839 (465,812) |
| 5 | Janssen Pharmaceutical K.K. | Novartis Pharma K.K. | Novartis Pharma K.K. |
| | 16,597,017 (152,546) | 20,211,538 (180,299) | 40,803,871 (369,567) |

Japanese yen (¥) were converted to US dollars ($) using the 2016 average monthly exchange rate of ¥108.8 per ($)1 and the 2017 average monthly exchange rate of ¥112.1 per ($)1.

Table 4 shows the results of the multiple negative binomial models for two-year combined payment monetary values among the CPG authors. Male CPG authors tended to receive larger payments than female ones (relatively monetary value (RMV) 0.52, 95% confidence interval (CI) 0.31 to 0.87). The professors who worked in universities or university hospitals were more likely to receive larger payments than non-professors (RMV 3.22, 95% CI 2.21 to 4.69). Furthermore, the greater the number of CPG the authors worked for, the larger payments they received (RMV 1.23, 95% CI 1.07 to 1.41).

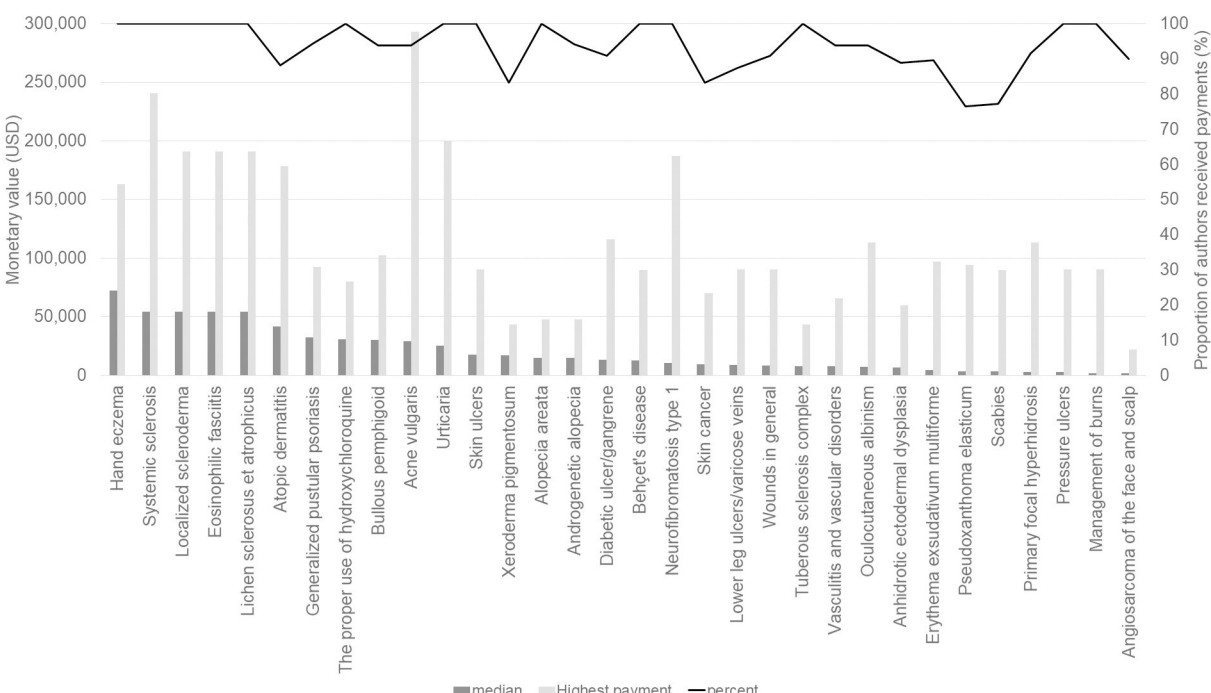

**Fig 1. Median and largest payment monetary values and proportion of the authors with at least one payment in each Clinical Practice Guideline.**

Table 5 describes the FCOIs for each CPG. We omitted 6 (18.8%) CPG in which the relevant authors had no FCOIs. Each CPG set the criteria governing the authors' disclosed of their FCOIs to the administrative office of the JDA (S4 Table). Overall, CPG authors were required to disclose all payments from Pharma when the total annual payment received exceeded 1 million Japanese yen (US$9,191) from any one company for speaking, writing and consulting work [14]. The minimum monetary value for the FCOI disclosure in each category was uniform among CPGs, while there were no rules about whether to disclose the FCOI status to the general public or not.

Of the 26 CPGs investigated, only the atopic dermatitis CPG and the acne vulgaris CPG published the FCOIs with individual author details. The atopic dermatitis CPG reported that 10 CPG authors out of a total of 17 submitted FCOIs, and detailed the purposes of the payments. Similarly, the acne vulgaris CPG reported that 7 CPG authors among 16 (43.8%) reported FCOIs. In contrast, 11 CPG (42.3%) disclosed the presence of FCOIs but without disclosing individual names and affiliations. There were actually no sections for FCOI disclosure in 13 CPG (50.0%). We found no changes in the extent of FCOI disclosures in subsequent annual publications (S3 Table).

We compared the FCOI disclosure of the two CPG with individual author details with the data issued by the individual Pharma. Although MSD K.K. ($9,213) and Mitsubishi Tanabe Pharma Corporation ($8,941) reported paying the authors for speaking between 2016 and 2017, respectively, two authors of the atopic CPG did not disclose any payments received from the companies. There were no such discrepancies in the acne vulgaris CPG.

## Discussion

In the analysis of 296 dermatology CPG authors in Japan, we revealed that 90.6% were reported by Pharma to have received at least one payment. We also found that the prevailing

**Table 3. Top five pharmaceutical companies in terms of the monetary values of payments to authors of the top five Clinical Practice Guidelines in the payment value.**

| Variable | Monetary value of payment, $ |
|---|---|
| **Hand eczema** | |
| Maruho Co. Ltd | 153,021 |
| Taiho Pharmaceutical Co. Ltd | 83,262 |
| Mitsubishi Tanabe Pharma Corporation | 75,589 |
| Kyowa Kirin Co. Ltd | 62,687 |
| Torii Pharmaceutical Co. Ltd | 49,069 |
| **Systemic sclerosis** | |
| Mitsubishi Tanabe Pharma Corporation | 126,495 |
| Maruho Co. Ltd | 66,337 |
| Bayer Yakuhin Ltd | 58,111 |
| Kyowa Kirin Co. Ltd | 53,532 |
| Janssen Pharmaceutical K.K. | 49,855 |
| **Localized scleroderma** | |
| Mitsubishi Tanabe Pharma Corporation | 102,229 |
| Maruho Co. Ltd | 87,614 |
| AbbVie GK | 50,819 |
| Kyowa Kirin Co. Ltd | 49,317 |
| Taiho Pharmaceutical Co. Ltd | 45,517 |
| **Lichen sclerosus et atrophicus** | |
| Mitsubishi Tanabe Pharma Corporation | 102,229 |
| Maruho Co. Ltd | 87,614 |
| AbbVie GK | 50,819 |
| Kyowa Kirin Co. Ltd | 49,317 |
| Taiho Pharmaceutical Co. Ltd | 45,517 |
| **Eosinophilic fasciitis** | |
| Mitsubishi Tanabe Pharma Corporation | 102,229 |
| Maruho Co. Ltd | 87,614 |
| AbbVie GK | 50,819 |
| Kyowa Kirin Co. Ltd | 49,317 |
| Taiho Pharmaceutical Co. Ltd | 45,517 |

Japanese yen (¥) were converted to US dollars ($) using the 2016 average monthly exchange rate of ¥108.8 per ($)1 and the 2017 average monthly exchange rate of ¥112.1 per ($)1.

FCOI disclosure systems were not effective in providing full transparency regarding financial relationships.

In the present study, we found that dermatology CPG authors received $13,300 (SD: $20,475) in the mean monetary value of payment per individual per year. Saito et al. reported the mean monetary values of payment per author per year were $10,565 (SD: $20,059) for oncology CPG authors and $11,568 (SD: $16,874) for orthopedic surgery professors in Japan [10,15]. Given that Executive Board members of the JDA received the second highest payments in the median values among those representing 18 basic medical fields in Japan [5], these figures suggested Japanese Pharma might focus more attention on dermatology than other medical fields.

In 2017, Checketts et al. reported that 40 of 49 authors (81.6%) of dermatology CPG by the American Academy of Dermatology received at least one payment from a US Pharma [12],

**Table 4. Negative binomial models for annual payment monetary values among Clinical Practice Guideline authors[a].**

| Variable | Relatively monetary values per year (95% confidence interval) |
|---|---|
| **Sex** | |
| Male | Ref. |
| Female | 0.52 (0.31–0.87) * |
| **Working affiliation** | |
| **Universities or university hospitals (non-professors)** | Ref. |
| **Universities or university hospitals (professors)** | 3.22 (2.21–4.69) *** |
| Other types of institutions | 1.16 (0.69–1.94) |
| **Number of guidelines (continuous variable)** | 1.23 (1.07–1.41)** |

[a] We excluded 8 non-physician authors

* $<0.05$

** $<0.01$

*** $<0.001$.

which is similar proportion to our study. However, the mean and median values for the authors in our study were smaller than those in the US (mean: $26,600 vs $83,703). One reason for this difference could be a difference in the categorization of the disclosed data between Japan and the US. The Open Payments Database in the US included payments related to food and beverage, travel and accommodation, gifts, and education. Moreover the payment data from Pharma in the US was mandatory rather than voluntary. In contrast, the Japanese data did not include such additional payments.

We found that the GI for the total payments was 0.69, and the top 10% of the Japanese CPG authors received about half of total payments ($3,529,929). These findings suggest a large disparity in payments among the authors, with the reported payments from Pharma concentrated on a small fraction of the authors, such as university professors. Universities and professors traditionally have been regarded as a symbol of authority in Japan. As repeatedly suggested in our previous works, university professors have a strong influence on practices and treatments in their clinical fields, and other physicians tend to follow a professors' decision of suitable treatment without question or criticism [13]. This result supports the idea that Pharma may be targeting and making payments to senior physicians who can influence of set clinical practice.

As shown in Table 3, several companies reported paying comparatively large amounts, perhaps reflecting the competition in the Japanese drug market. The market for biologic therapies has expanded in recent years and the market scale is now about US$8.22 billion in Japan. Actually, in February 2015, Japan's Maruho Co Ltd and Novartis Pharma K.K. launched secukinumab (COSENTYX, approval year 2014) for psoriasis and ankylosing spondylitis, which was one of the top-selling products of Maruho. Other Pharma have also developed various biologic therapies, such as brodalumab (LUMICEF, approval year 2016) for psoriasis and psoriatic erythroderma from Kyowa Kirin Co Ltd; adalimumab (HUMIRA, approval year 2016) for psoriasis, ankylosing spondylitis and rheumatic arthritis from AbbVie GK; infliximab (REMICADE, approval year 2002) for psoriasis, Behçet's disease, ankylosing spondylitis and rheumatic arthritis from Mitsubishi Tanabe Pharma Corporation; ustekinumab (STELARA, approval year 2011) and guselkmab (TREMFYA, approval year 2018) for psoriasis from Janssen Pharmaceutical K.K. The use of adalimumab and infliximab were approved for generalized pustular psoriasis, localized scleroderma and eosinophilic fasciitis in the 2016 dermatology CPGs, and

**Table 5. Characteristics of Clinical Practice Guidelines considered in this study.**

| Topics of guideline | Date of publication | Extents of financial Conflict of Interest disclosure |
|---|---|---|
| Atopic dermatitis | November 20, 2018 | Disclosure with individual details |
| Urticaria | November 20, 2018 | Disclosure with aggregated data |
| Behçet's disease | September 20, 2018 | No disclosure |
| Hand eczema | March 20, 2018 | No disclosure |
| Anhidrotic ectodermal dysplasia | February 20, 2018 | No disclosure |
| Tuberous sclerosis complex | January 20, 2018 | No disclosure |
| Neurofibromatosis type 1 | January 20, 2018 | Omitted as there reported to be no financial conflicts of interest |
| Alopecia areata | December 20, 2017 | Omitted as there reported to be no financial conflicts of interest |
| Androgenetic alopecia | December 20, 2017 | Disclosure with an aggregated data |
| Pseudoxanthoma elasticum | October 20, 2017 | Omitted as there reported to be no financial conflicts of interest |
| Lower leg ulcers/varicose veins | September 20, 2017 | Disclosure with an aggregated data |
| Management of burns | September 20, 2017 | Disclosure with an aggregated data |
| Pressure ulcers | August 20, 2017 | Disclosure with an aggregated data |
| Diabetic ulcer/gangrene | August 20, 2017 | Disclosure with an aggregated data |
| Skin ulcers associated with connective tissue disease/vasculitis. | August 20, 2017 | Disclosure with an aggregated data |
| Wounds in general | July 20, 2017 | Disclosure with an aggregated data |
| Bullous pemphigoid | June 20, 2017 | No disclosure |
| Acne vulgaris | May 20, 2017 | Disclosure with individual details |
| Vasculitis and vascular disorders | March 20, 2017 | No disclosure |
| Oculocutaneous albinism | February 20, 2017 | No disclosure |
| Lichen sclerosus et atrophicus | November 20, 2016 | No disclosure |
| Eosinophilic fasciitis | November 20, 2016 | No disclosure |
| Localized scleroderma | October 20, 2016 | No disclosure |
| Systemic sclerosis | September 20, 2016 | No disclosure |
| Erythema exsudativum multiforme major, Stevens-Johnson syndrome and toxic epidermal necrolysis | August 20, 2016 | Omitted as there reported to be no financial conflicts of interest |
| Generalized pustular psoriasis | November 20, 2015 | Disclosure with an aggregated data |
| Scabies | October 20, 2015 | Omitted as there reported to be no financial conflicts of interest |
| Xeroderma pigmentosum | October 20, 2015 | No disclosure |
| The proper use of hydroxychloroquine | October 20, 2015 | Disclosure with an aggregated data |
| Skin cancer | October 20, 2015 | Omitted as there reported to be no financial conflicts of interest |
| Angiosarcoma of the face and scalp | September 20, 2015 | Disclosure with an aggregated data |
| Primary focal hyperhidrosis | June 20, 2015 | No disclosure |

some biologic therapies have been used for pustular psoriasis, localized scleroderma and pla-que psoriasis. This field may be an important target for promotional activities from numerous rival companies with significant promotional funds, leading to the large payments reportedly paid to dermatology CPG authors.

Our study found the proportion of female CPG authors was significantly lower than that of male authors and industry payments to female CPG authors was also significantly lower. Kathyrn et al. reported Pharma tended to make more payments to male physicians than to female ones in the US [16,17]. Further, in Japan, similar findings were observed in the case of certified oncologists [13], and our findings were consistent with the previous studies. Although there are 1.25 times more male dermatologists than female dermatologists (3189 male derma-tologists and 2543 female dermatologist) in Japan [18], the lower proportion of female CPG authors and lower Pharma payments to female CPG authors could not be explained merely by the discrepancy in numbers alone. Dermatology is one of the most attractive medical special-ties for females in Japan, as shown by the fact that, overall, only 21% of Japanese physicians are female whereas the figure in dermatology is more than double of that (44%). Dermatology offers better and more amenable job opportunities for women, such as working hours (i.e. a normal 9–5 working day with little overtime), less exhausting work, few if any invasive proce-dures and far less prospect of being sued for malpractice [19]. This allows women to, as far as possible, follow the society-driven role of being a homemaker, as well as maintain their employment as a physician. Nonetheless the status of females in Japan has been traditionally much lower compared with male counterparts with little recent tangible improvement [20,21]. There are long-standing and profound prejudices regarding females in the male-dominated and patriarchal Japanese medical community. The nation discriminates against female physi-cians, with Japan's total of female physicians being the lowest among industrialized nations. As a recent illustration of the degree of discrimination against females in the medical field in Japan, in 2018, it became evident that many medical schools were suppressing the number of female medical students by manipulating entrance examination scores to ensure that many women could not gain entrance to medical schools while males with lower scores were accepted [22]. The lack of gender equality in all aspects of life in Japanese society is manifest in the prevailing belief by the male-dominated hierarchy that a woman's role is to get married and become a housewife and raise children [23–25]. Consequently, in the medical field, it is assumed that any women qualifying as physicians will reasonably quickly relinquish their posts to marry, commit to domestic duties and raise their children, resulting in a waste of resources needed to educate and train them and difficulties in replacing them when they quit. Partially supported by these prejudices against females, in Japan's male dominated society, male physicians usually hold higher academic positions, such as directors of hospital and chairpersons of CPG committees. Therefore, Pharma may concentrate their activities on male CPG authors who are in influential positions rather than on female ones.

Of the 32 dermatology CPGs, the hand eczema CPG authors received largest payments ($72,143) in median values. Especially Maruho Co. Ltd made largest payments to the hand eczema CPG authors, and contributed 11.2% of total payments ($153,021). Maruho Co. Ltd sells heparinoid (Hirudoid, approved in 2008), which accounted for 64.0% of its sales in 2017, and the hand eczema CPG recommended using moisturizing agents including heparinoid. The systemic sclerosis CPG ($54,398) was the second highest median payments. The systemic sclerosis CPG covered criteria and treatment for not only dermal sclerosis but also related gas-trointestinal disease, interstitial lung disease, and renal disease. Thus, there are more drugs involved in the systemic sclerosis CPG than in other dermatology CPGs. For Pharma, systemic sclerosis CPG could have been a major target for increasing their sales income, especially so in view of there being a relatively small number of CPG authors and a large number of drugs

related to systemic sclerosis treatment options. Indeed, each Pharma listed among the top five companies making payments in relation to systemic sclerosis CPG authors has therapeutic products recommended for use in the systemic sclerosis CPG.

We found that the extent of the FCOI disclosure in each CPG was inaccurate and lacking. JDA's FCOI guideline is quite weak, meaning all authors receiving 1 million Japanese yen or less (US$9191) per payment have no obligation to designate such payments as an FCOI or make any such disclosure [14]. Further, as the disclosure depends on self-declaration, there is no system for monitoring the accuracy of FCOI disclosures in CPGs. There is also no mandatory policy, policing or punishment with respect to Pharma and the payment data they report. Ideally, CPG authors should declare the full amount of payments they receive related to CPG development, as in the American Academy of Dermatology policy [26]. Until a new FCOI disclosure mechanism is devised and implemented we will be unable to compare the situation in Japan with that in other countries. The inability to compare data between nations will hamper the introduction of a cohesive policy to improve transparency and to enable an improvement in trust that physicians will be making treatments practice decisions based on sound medical evidence rather than being influenced by financial 'incentives' or other corrupt or unethical activities emanating from Pharma.

## Clinical implications and future perspectives

Although Pharma have contributed to advancement of medicine, CPG authors should be free from influence of the Pharma and each medical society should minimalize interaction with the pharmaceutical industry [27]. We suggest several solutions for more transparent and credible CPGs. First, the JDA should set more rigorous FCOI disclosure criteria, such as CPG authors declaring and disclosing full amount of receipts in their CPG disclosures, as is the case with the American Academy of Dermatology. In addition, the JDA should prohibit greater than 50% of all CPG authors from receiving speaking and consulting fees for the duration of the CPG development period and up to one year following the announcement of a new guideline, as per the American Urological Association [28]. Further, healthcare professionals with a proven profound financial relationship with Pharma should be excluded from being a CPG author, as Saito et al. suggested in a previous study [10]. To police and enforce these policies, independent auditing organizations, which are free from pharmaceutical industry connections, will be needed. Second, disclosure of the current payment data in Japan should be revisited, as the format of payment data is not user-friendly, differing between the companies involved. The types and amounts of payments need to be standardized in Japan as well as globally and made compulsory, with full disclosures needing to be made with regard to the reasons for the payments. Third, Healthcare professionals who disagree with the data published by Pharma should have a simple, low-cost mechanism to settle any disputes, with the same system available to Pharma should it be needed. Fourth, to confirm the accuracy of FCOI disclosures made by CPG authors, we suggest the development of a new official payment database in Japan, similar to the world's only legally binding Open Payment database in the US [29]. So far, it has been controversial and not brought about any positive change, has not functioned well, and has not accomplished its desired goal [30]. Still, In Japan and elsewhere where the payments databases are controlled by trade associations, the notoriously secretive payments of Pharma cannot be independently verified in any respect. the recent international movement to examine the interactions between Pharma and physicians has, at least, focused attention on the possibly corrupt and unethical financial relationships between physicians and Pharma, including in Japan. Finally, given that FCOI may occur because of Pharma strategies to maximize the own benefits, to nationalize all Pharma is one of the possible solutions for managing any kind of FCOI. Nationalized Pharma

would prioritize the production of public health goods, not the pursuit of profits, and they would not need to make payments to physicians, apart from to promote the research and development of products to improve public health [31].

## Limitations

There are several limitations in this study. First, there may also be inaccuracies in the payment database and details of CPG authors. Many Pharma disclosed their payment data in varying formats, which were not uniform and easily comparable. Therefore, we needed to identify all payments, names and affiliations manually. Although the accuracy of the data was carefully and repeatedly reviewed, the payment database might include human errors in data entry. In addition, the current mechanism in Japan has no way of dealing with any discrepancies or disputes about payments raised by either Pharma or individual physicians, although our team have always revisited an accuracy of our handling of disclosed data, upon reasonable inquiry to the Money Database from concerned individuals in the published data and have made a consultation to the company that disclosed the relevant data and fixed it when necessary. Further, because the CPG authors failed to confirm the presence of their financial arrangement with Pharma, the results of the present study might include errors, plus amounts specified may, potentially, have exceeded the amounts that physicians actually received. However, this is not only the case in Japan, as the payment databases coordinated by the Association of the British Pharmaceutical Industry in the UK and Medicines Australia in Australia also apparently having no formal dispute system. According to the study in the US reported by Feng et al, 7 dermatologists out of 8333 dermatologists disputed 36 payments ($61,278.47) out of a total of 208,613 payments totaling $34,810,661.57 [32]. We estimate that the effect of disputes would be small enough in our study. Second, the present research payment data were limited, as Pharma were asked to report payments only for lecturing, writing and consulting work, not for food and beverages, stock holdings, travel and accommodation, gifts, education, research work, etc. Consequently, a comprehensive picture of the actual financial relationship between CPG authors and industrial companies was not possible. Unfortunately 2018 payment data could not be included in our analysis because we were still compiling and cleaning up the 2018 data prior to integrating it into our database. However, as the payment patterns from Pharma between 2016 and 2017 were similar, the effect of adding 2018 payment data would be small. Consequently, we restricted our study and only used the fully processed payment data reported for 2016 and 2017 at time of the study initiation.

## Conclusions

In Japan, most authors of dermatology CPGs reportedly received payments from Pharma. However, the extent of the FCOI disclosure of these authors, when they were required and/or made were far from uniform, accurate or adequate. Moreover, the criteria and rules governing FCOI disclosure were also inadequate and not fit for purpose. Stricter criteria for FCOI disclosure need to be created, imposed and policed, along with mandatory disclosures of all relevant payments from Pharma to any and all physicians, in order to allay all possible claims or perceptions of corruption and unethical behavior with regard to medical practice. The paramount goal must always be the safety and wellbeing of patients rather than the pursuit of profits on the part of Pharma or practicing physicians.

## Supporting information

**S1 Fig. Distribution of payment monetary values per Clinical Practice Guideline authors.** (TIF)

**S1 Table. Characteristics of Clinical Practice Guidelines considered in this study.**
(DOCX)

**S2 Table. Payment characteristics of authors for each dermatology Clinical Practice Guideline.** Japanese yen (¥) were converted to US dollars ($) using the 2016 average monthly exchange rate of ¥108.8 per ($)1 and the 2017 average monthly exchange rate of ¥112.1 per ($) 1.
(DOCX)

**S3 Table. Extents of financial conflict of interest disclosure in each Clinical Practice Guideline by publication year.**
(DOCX)

**S4 Table. Criteria for the financial Conflict of Interest disclosure to the administrative office of the Japanese Dermatological Association.** Japanese yen (¥) were converted to US dollars ($) using the 2016 average monthly exchange rate of ¥108.8 per ($)1.
(DOCX)

**S1 File.**
(XLSX)

## Acknowledgments

The authors thank the Waseda Chronicle for providing payments data, Professor Andy Crump for constructive opinion, and Ms. Erika Yamashita for organizing payment data.

## Author Contributions

**Conceptualization:** Anju Murayama, Akihiko Ozaki, Hiroaki Saito, Toyoaki Sawano, Yuki Shimada, Kana Yamamoto, Yosuke Suzuki, Tetsuya Tanimoto.

**Data curation:** Anju Murayama, Akihiko Ozaki.

**Formal analysis:** Anju Murayama, Akihiko Ozaki, Hiroaki Saito, Toyoaki Sawano, Yuki Shimada, Kana Yamamoto, Yosuke Suzuki, Tetsuya Tanimoto.

**Funding acquisition:** Akihiko Ozaki.

**Investigation:** Anju Murayama, Akihiko Ozaki, Hiroaki Saito, Tetsuya Tanimoto.

**Methodology:** Anju Murayama, Akihiko Ozaki, Hiroaki Saito, Toyoaki Sawano, Yuki Shimada, Kana Yamamoto, Yosuke Suzuki, Tetsuya Tanimoto.

**Project administration:** Anju Murayama, Akihiko Ozaki, Tetsuya Tanimoto.

**Resources:** Anju Murayama.

**Software:** Anju Murayama.

**Supervision:** Akihiko Ozaki, Tetsuya Tanimoto.

**Validation:** Anju Murayama, Hiroaki Saito, Toyoaki Sawano, Yuki Shimada, Kana Yamamoto, Yosuke Suzuki, Tetsuya Tanimoto.

**Visualization:** Anju Murayama, Hiroaki Saito.

**Writing – original draft:** Anju Murayama, Akihiko Ozaki, Tetsuya Tanimoto.

**Writing – review & editing:** Anju Murayama, Akihiko Ozaki, Tetsuya Tanimoto.

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
