## [Decision Letter · Decision Letter 0]

4 Jun 2020

PONE-D-20-12296

Pharmaceutical company payments to dermatology Clinical Practice Guideline authors in Japan in 2016

PLOS ONE

Dear Dr. Murayama,

Thank you for submitting your manuscript to PLOS ONE. After careful consideration, we feel that it has merit but does not fully meet PLOS ONE’s publication criteria as it currently stands. Therefore, we invite you to submit a revised version of the manuscript that addresses the points raised during the review process.

In particular the methodological issues raised by the reviewers should be carefully considered. 

We look forward to receiving your revised manuscript.

Kind regards,

Tim Mathes

Academic Editor

PLOS ONE

Journal Requirements:

"This study was supported from the Medical Governance Research Institute, Ain

Pharmaciez and the Waseda Chronicle.

The funders had no role in study design, data collection and analysis, decision to

publish, or preparation of the manuscript."

We note that you received funding from a commercial source: "Ain Pharmaciez"

Reviewers' comments:

Reviewer's Responses to Questions

**Comments to the Author**

1. Is the manuscript technically sound, and do the data support the conclusions?

Reviewer #1: Yes

Reviewer #2: No

2. Has the statistical analysis been performed appropriately and rigorously? 

Reviewer #1: Yes

Reviewer #2: Yes

3. Have the authors made all data underlying the findings in their manuscript fully available?

Reviewer #1: Yes

Reviewer #2: Yes

4. Is the manuscript presented in an intelligible fashion and written in standard English?

Reviewer #1: Yes

Reviewer #2: Yes

5. Review Comments to the Author

Reviewer #1: Introduction:

Very well written, I especially enjoyed the section explaining WHY dermatology may receive more than other other specialties. The authors could add their hypothesis to the bottom of the last paragraph IF they had an established hypothesis prior to conducting the study.

Methodology:

I think this study needs a more thorough explanation of the methods of locating the CPG authors relationships with industry. For example, in the USA we would use Open Payments. What database did the authors use? How did they extract from the database? How did they know they found the correct person? How did they solve disputes? And further explanations regarding this process. I know the authors cited a previous study they did, but I feel for such a touchy subject they should have a more thorough methodology section.

The rest of the methodology is adequate.

Results:

Very well reported. Nothing to add.

Discussion:

The citation on line 311 should be "Checketts et al." not "Jake et al."

The paragraph regarding your findings based on gender is very interesting and adds to the literature.

I think this paper is missing a paragraph before the limitations in which the authors provide actionable recommendations for improving the presence of FCOIs among CPG authors in Japan. This would significantly add to this work.

The rest of the discussion is great, well written, and provides interesting information that differs from other studies on this subject.

Reviewer #2: The focus of this study are authors of dermatology clinical practice guidelines (CPGs). As the emphasis of the study is within the field of dermatology, it would be worthwhile to know how many of the authors included in the analysis were non-dermatologists. Several of the CPGs listed could include other subspecialists. For example, for the following CPGs: “Management of burns”, “Pressure ulcers”, “Neurofibromatosis”, “Bechet’s disease”, and “Angiosarcoma of the face and scalp”, other subspecialists (vascular surgeons, rheumatologists, neurologists, pediatricians, oncologists) could have contributed. This is particularly important to know as the framing of the introduction and discussion of the study is centered on potential financial conflicts of interests from high dollar biological therapies to dermatologists specifically.

While this was addressed as a limitation of the study in the discussion, the source of the data is limited. It is difficult to draw significant conclusions when the source data is not reliable. The self reporting of payments by the pharmaceutical companies is not standardized. Payments could far exceed what is actually being reported.

While the statistical analysis is sound, the results don’t readily support the proposed conclusions fully. Again, particular attention is paid in the introduction and discussion of the paper to biological therapies for psoriasis and other conditions. However, the highest payments to authors were not for psoriasis but rather other less common dermatologic conditions that are not commonly treated with biologics. With the exception of hand eczema, the other top 5 CPGs in payment value aren’t traditionally treated with biological therapies.

It should be noted that the payments data is from 2016 but well over half of the CPGs were published after 2016. The source of payment data (http://www.jpma.or.jp/tomeisei/guideline/) has payment data from 2017 and 2018. Why wasn’t this payment data included in the study given that many of the CPGs were published after 2016?

In discussing the sex differences between male and female CPG authors, it would be worthwhile to discuss the demographics of practitioners in Japan. Are more men receiving payments because there are more male physicians in Japan to begin with? Also is there any data or studies from other specialty groups in Japan by which to compare the dermatologist payments?

6. PLOS authors have the option to publish the peer review history of their article (what does this mean?). If published, this will include your full peer review and any attached files.

Reviewer #1: Yes: Jake Checketts

Reviewer #2: No

---

## [Author Response · Author response to Decision Letter 0]

2 Aug 2020

August 3, 2020

Dr. Tim Mathes

Academic Editor

PLoS ONE

Dear Dr. Mathes

Pharmaceutical company payments to dermatology Clinical Practice Guideline authors in Japan.

Please find enclosed the revised manuscript in Microsoft Word format, which my co-authors and I are pleased to re-submit for consideration for publication in the PLoS ONE journal.

We have taken the comments and suggestions of the Peer Reviewers into account and revised the text accordingly. We are also submitting a separate Response Letter in which we present itemized explanations of our responses to the points raised by the Peer Reviewers. 

Also, we emended the article name from “Pharmaceutical company payments to dermatology Clinical Practice Guideline authors in Japan in 2016” to “Pharmaceutical company payments to dermatology Clinical Practice Guideline authors in Japan” due to adding 2017 payment data. Further we added Competing Interests Statement in both of this cover letter and the revised manuscript. Our institute, Medical Governance Research Institute, received donation from Ain Pharmaciez. Also Dr. Saito received personal fees from TAIHO Pharmaceutical Co., Ltd. outside the scope of the submitted work. Drs. Ozaki and Tanimoto received personal fees from Medical Network Systems outside the scope of the submitted work. This donation from Ain Pharmaceiz does not alter our adherence to PLoS ONE policies on sharing data and materials. 

We hope that this revision will meet with the standard of approval for publication in PLoS ONE, and we look forward to hearing from you in the near future.

Sincerely,

Anju Murayama

Medical Governance Research Institute

2-12-13 Takanawa, Minato-ku, Tokyo, Japan

1087505

Telephone: 81-90-6321-6996

Email: ange21tera@gmail.com  

Pharmaceutical company payments to dermatology Clinical Practice Guideline authors in Japan.

Response Letter

Reviewer #1 

Introduction

The authors could add their hypothesis to the bottom of the last paragraph IF they had an established hypothesis prior to conducting the study.

 Reply:

 Thank you for your comments. We have added text outlining our concept as follows:

 (Line 77 in the revised manuscript)

 “Pharma making payments may be aggressively and unethically promoting the sale and use of their drugs. Therefore, we hypothesized that, by examining dermatology CPG authors in Japan who received financial payments from Pharma, we could elucidate whether the amount of money received would be higher among those authoring a larger number of CPGs or in cases where the CPGs recommended the use of drugs newly marketed by the companies making the payments.”

Methodology:

I think this study needs a more thorough explanation of the methods of locating the CPG authors relationships with industry. For example, in the USA we would use Open Payments. What database did the authors use? 

Reply: 

We have added a more detailed description about how our payment database was created, as follows: 

(Line 107 in the revised manuscript)

“Payment data were published on the website of each company which was, at the time, a member of the Japan Pharmaceutical Manufacturing Association (JPMA). We collected the payment data from the 78 and 75 companies which belonged to the JPMA in 2016 and in 2017, respectively, as in our previous study [13]. 

Using the collected data, we generated a unified single database, as follows. First, because no data were published in the form of a spreadsheet or in any standardized fashion, data with character codes were converted into a spreadsheet format. Second, data with no character code were converted into text files using an optical character reader (Yomitori kakumei, version 15; Panasonic Solution Technologies Company, Ltd, Tokyo, Japan). Third, for data protected against any form of reproduction, we used FullShot, version 10 software (Inbit Inc, California, USA) to scan the data and convert the resultant images into text files. Finally, we confirmed that the transformed data were accurately converted by comparing them with the original data. Our database included the names of all individual physicians, their primary affiliated institutions, the amounts of payments made by Pharma, and the forms for the payments. The form of payments used was limited, being categorized into the following three types: payment for lectures, payment for authoring, editing, etc., and consulting fees. The data did not include research payments, meal and the benefits, because the Pharma concerned did not report these as separate, identifiable payments [3].”

How did they extract from the database? 

Reply: 

In response to your request for further information, we have revised the text and added a more detailed description about how payment data was extracted; 

(Line 128 in the revised manuscript)

“From the payment database, we extracted payment data reported by each company as having been paid to each individual physician by matching individual names using the Excel function “iferror” and “vlookup”.”

How did they know they found the correct person? 

Reply: 

We have added more detailed description as to how each individual was identified; 

(Line 130 in the revised manuscript)

“For each person named in the database we checked to find and remove any and all duplicates. For each name included, we also identified the work affiliation specified by the company making a payment and the area and/or specialty of the individuals concerned. We also visited the websites of their main places of work and, where possible, found biographies and photos of the individuals concerned to confirm the identity of the CPG authors.”

How did they solve disputes? 

Reply: 

There is no formal dispute scheme in Japan, although our team have always revisited an accuracy of our handling of disclosed data, upon reasonable inquiry to the Money Database from concerned individuals in the published data and have made a consultation to the company that disclosed the relevant data and fixed it when necessary. In any case, as is known from similar payment databases around the world, the matter of non-agreement about payments cannot easily be resolved. This is the case in all of the world’s non-legally binding sites (such as that in Japan) as well as the legally-binding Open Payments database in the United States, where any dispute about payments made and/or received will usually end up in a court of law. Countries such as the UK and Australia, which also use similar payment databases, face the same situation. Given the study reporting financial relationships between dermatologists and pharmaceutical companies by Feng et al, we think the effect of disputes would be small, and we revised limitation section, as follows.

(Line 448 in the revised manuscript)

“In addition, the current mechanism in Japan has no way of dealing with any discrepancies or disputes about payments raised by either Pharma or individual physicians, although our team have always revisited an accuracy of our handling of disclosed data, upon reasonable inquiry to the Money Database from concerned individuals in the published data and have made a consultation to the company that disclosed the relevant data and fixed it when necessary. Further, because the CPG authors failed to confirm the presence of their financial arrangement with Pharma, the results of the present study might include errors, plus amounts specified may, potentially, have exceeded the amounts that physicians actually received. However, this is not only the case in Japan, as the payment databases coordinated by the Association of the British Pharmaceutical Industry in the UK and Medicines Australia in Australia also apparently having no formal dispute system. According to the study in the US reported by Feng et al, 7 dermatologists out of 8333 dermatologists disputed 36 payments ($61,278.47) out of a total of 208,613 payments totaling $34,810,661.57 [32].”

And further explanations regarding this process. I know the authors cited a previous study they did, but I feel for such a touchy subject they should have a more thorough methodology section.

Reply: 

Following your comment and suggestion, we have added full details of descriptions about process of finalizing our payment data, as follows:

(Line 105 in the revised manuscript)

“Payment data were published on the website of each company which was, at the time, a member of the Japan Pharmaceutical Manufacturing Association (JPMA). We collected the payment data from the 78 and 75 companies which belonged to the JPMA in 2016 and in 2017, respectively, as in our previous study [13]. 

Using the collected data, we generated a unified single database, as follows. First, because no data were published in the form of a spreadsheet or in any standardized fashion, data with character codes were converted into a spreadsheet format. Second, data with no character code were converted into text files using an optical character reader (Yomitori kakumei, version 15; Panasonic Solution Technologies Company, Ltd, Tokyo, Japan). Third, for data protected against any form of reproduction, we used FullShot, version 10 software (Inbit Inc, California, USA) to scan the data and convert the resultant images into text files. Finally, we confirmed that the transformed data were accurately converted by comparing them with the original data. Our database included the names of all individual physicians, their primary affiliated institutions, the amounts of payments made by Pharma, and the forms for the payments. The form of payments used was limited, being categorized into the following three types: payment for lectures, payment for authoring, editing, etc., and consulting fees. The data did not include research payments, meal and the benefits, because the Pharma concerned did not report these as separate, identifiable payments [3]. 

From the payment database, we extracted payment data reported by each company as having been paid to each individual physician by matching individual names using the Excel function “iferror” and “vlookup”. For each person named in the database we checked to find and remove any and all duplicates. For each person named in the database we checked to find and remove any and all duplicates. For each name included, we also identified the work affiliation specified by the company making a payment and the area and/or specialty of the person concerned, We also visited the websites of their main places of work and, where possible, found biographies and photos of the individuals concerned to help confirm the identity of the CPG authors.

We used data on physicians’ names, their main work affiliations, the amount of payments, payment formats, and the total number of payments from our payment database. The form of payments was categorized into 3 types: lecturing, writing work, and consulting fees.”

Discussion:

The citation on line 311 should be "Checketts et al." not "Jake et al."

Reply: 

Thank you for your observation, we have revised the citation information on line 291. 

The paragraph regarding your findings based on gender is very interesting and adds to the literature.

Reply:

In the revised manuscript, we have added further information to facilitate a better understanding of problems surrounding female physicians in Japan as follows: 

(Line 351 in the revised manuscript)

“There are long-standing and profound prejudices regarding females in the male-dominated and patriarchal Japanese medical community. The nation discriminates against female physicians, with Japan’s total of female physicians being the lowest among industrialized nations. As a recent illustration of the degree of discrimination against females in the medical field in Japan, in 2018, it became evident that many medical schools were suppressing the number of female medical students by manipulating entrance examination scores to ensure that many women could not gain entrance to medical schools while males with lower scores were accepted [22]. The lack of gender equality in all aspects of life in Japanese society is manifest in the prevailing belief by the male-dominated hierarchy that a woman’s role is to get married and become a housewife and raise children [23-25]. Consequently, in the medical field, it is assumed that any women qualifying as physicians will reasonably quickly relinquish their posts to marry, commit to domestic duties and raise their children, resulting in a waste of resources needed to educate and train them and difficulties in replacing them when they quit.”

I think this paper is missing a paragraph before the limitations in which the authors provide actionable recommendations for improving the presence of FCOIs among CPG authors in Japan. This would significantly add to this work.

Reply: 

We have revised the manuscript, as follows.

(Line 404 in the revised manuscript)

“Although Pharma have contributed to advancement of medicine, CPG authors should be free from influence of the Pharma and each medical society should minimalize interaction with the pharmaceutical industry [27]. We suggest several solutions for more transparent and credible CPGs. First, the JDA should set more rigorous FCOI disclosure criteria, such as CPG authors declaring and disclosing full amount of receipts in their CPG disclosures, as is the case with the American Academy of Dermatology. In addition, the JDA should prohibit greater than 50% of all CPG authors from receiving speaking and consulting fees for the duration of the CPG development period and up to one year following the announcement of a new guideline, as per the American Urological Association [28]. Further, healthcare professionals with a proven profound financial relationship with Pharma should be excluded from being a CPG author, as Saito et al. suggested in a previous study [10]. To police and enforce these policies, independent auditing organizations, which are free from pharmaceutical industry connections, will be needed. Second, disclosure of the current payment data in Japan should be revisited, as the format of payment data is not user-friendly, differing between the companies involved. The types and amounts of payments need to be standardized in Japan as well as globally and made compulsory, with full disclosures needing to be made with regard to the reasons for the payments. Third, Healthcare professionals who disagree with the data published by Pharma should have a simple, low-cost mechanism to settle any disputes, with the same system available to Pharma should it be needed. Fourth, to confirm the accuracy of FCOI disclosures made by CPG authors, we suggest the development of a new official payment database in Japan, similar to the world’s only legally binding Open Payment database in the US [29]. So far, it has been controversial and not brought about any positive change, has not functioned well, and has not accomplished its desired goal [30]. Still, In Japan and elsewhere where the payments databases are controlled by trade associations, the notoriously secretive payments of Pharma cannot be independently verified in any respect. the recent international movement to examine the interactions between Pharma and physicians has, at least, focused attention on the possibly corrupt and unethical financial relationships between physicians and Pharma, including in Japan. Finally, given that FCOI may occur because of Pharma strategies to maximize the own benefits, to nationalize all Pharma is one of the possible solutions for managing any kind of FCOI. Nationalized Pharma would prioritize the production of public health goods, not the pursuit of profits, and they would not need to make payments to physicians, apart from to promote the research and development of products to improve public health [31].”

Reviewer #2: 

As the emphasis of the study is within the field of dermatology, it would be worthwhile to know how many of the authors included in the analysis were non-dermatologists. Several of the CPGs listed could include other subspecialists. For example, for the following CPGs: “Management of burns”, “Pressure ulcers”, “Neurofibromatosis”, “Bechet’s disease”, and “Angiosarcoma of the face and scalp”, other subspecialists (vascular surgeons, rheumatologists, neurologists, pediatricians, oncologists) could have contributed.

Reply: 

We thank you for your observations and insightful comments in this regard. We have added information on the number of dermatologists and other specialists in Table 1, and the number of dermatologists and other specialists by each CPG in Table S1. 

While this was addressed as a limitation of the study in the discussion, the source of the data is limited. It is difficult to draw significant conclusions when the source data is not reliable. The self reporting of payments by the pharmaceutical companies is not standardized. Payments could far exceed what is actually being reported.

Reply: 

We appreciate your valuable comments on this issue. In Japan, there is no other choice but to use self-reporting payment data from pharmaceutical companies. Unlike the US, there is no way to formally dispute payments reported by pharmaceutical companies in Japan, though our team have always revisited an accuracy of our handling of disclosed data, upon reasonable inquiry to the Money Database from concerned individuals in the published data and have made a consultation to the company that disclosed the relevant data and fixed it when necessary. In this context, there might be some discrepancies between the reported payment from pharmaceutical companies and actual receipt of the payment, as reviewer 2 mentioned. Now we have revised the manuscript as follows. 

(Line 448 in the revised manuscript)

“In addition, the current mechanism in Japan has no way of dealing with any discrepancies or disputes about payments raised by either Pharma or individual physicians, although our team have always revisited an accuracy of our handling of disclosed data, upon reasonable inquiry to the Money Database from concerned individuals in the published data and have made a consultation to the company that disclosed the relevant data and fixed it when necessary. Further, because the CPG authors failed to confirm the presence of their financial arrangement with Pharma, the results of the present study might include errors, plus amounts specified may, potentially, have exceeded the amounts that physicians actually received. However, this is not only the case in Japan, with the payment databases coordinated by the Association of the British Pharmaceutical Industry in the UK and Medicines Australia in Australia also apparently having no formal dispute system. In any case, according to the study reported by Feng et al, 7 dermatologists out of 8333 dermatologists disputed 36 payments ($61,278.47) out of a total of 208,613 payments totaling $34,810,661.57[32]. We estimate that the effect of disputes would be small enough in our study.” 

While the statistical analysis is sound, the results don’t readily support the proposed conclusions fully. Again, particular attention is paid in the introduction and discussion of the paper to biological therapies for psoriasis and other conditions. However, the highest payments to authors were not for psoriasis but rather other less common dermatologic conditions that are not commonly treated with biologics. With the exception of hand eczema, the other top 5 CPGs in payment value aren’t traditionally treated with biological therapies.

Reply: 

CPGs for diseases treated with biological and other expensive therapies, such as psoriasis and melanoma, were not included in this study because these CPGs were published after the end of 2018. By adding 2017 payment data, in the revised manuscript, a CPG for hand eczema was ranked first in the Top 5 CPGs in terms of payment monetary value. Most of the payments (79.2%) were categorized as speaking fees, and it can be assumed that some pharmaceutical companies hold conferences for promoting new drugs including biological therapies. Since the payment data does not include the specific purpose of the payment, we cannot confirm that conferences held focused on new drugs. However, we considered that the relationship between pharmaceutical companies that paid larger amounts and sold relevant drugs, including biological therapies, was of particular significance in supporting our basic hypothesis. 

It should be noted that the payments data is from 2016 but well over half of the CPGs were published after 2016. The source of payment data (http://www.jpma.or.jp/tomeisei/guideline/) has payment data from 2017 and 2018. Why wasn’t this payment data included in the study given that many of the CPGs were published after 2016?

Reply: 

We understand your observation and reservation about the most current data not being used in our analysis. We have revised the manuscript and have now included 2017 payment data, which is the latest ready payment data in Japan, and revised the analysis using both 2016 and 2017 payment data. The 2018 payment data was not available because we are still integrating and cleaning up the 2018 payment data. However, the trend of payments from Pharma did not change so much between 2016 and 2017, and we think the effect of adding 2018 payment data would be small enough. To reflect this situation, we have revised the manuscript as follows.

(Line 467 in the revised manuscript)

“Unfortunately 2018 payment data could not be included in our analysis because we were still compiling and cleaning up the 2018 data prior to integrating it into our database. However, as the payment patterns from Pharma between 2016 and 2017 were similar, the effect of adding 2018 payment data would be small. Consequently, we restricted our study and only used the fully processed payment data reported for 2016 and 2017 at time of the study initiation.”

In discussing the sex differences between male and female CPG authors, it would be worthwhile to discuss the demographics of practitioners in Japan. Are more men receiving payments because there are more male physicians in Japan to begin with?

Reply: 

Thank you for your observations and comments on the gender issues in Japan. Our finding of payment differences between male and female CPG authors was based solely on the payments reported by Pharma to individual authors. This allowed us to explore whether there were any gender differences involved. Actually, in the dermatology field there are more male physicians than female physicians in Japan (3189 males versus 2543 females). However, the greater number of males does not fully explain our findings. Due to the prevailing gender inequality in Japan, male physicians are almost always occupying higher, more influential positions than female physicians. It is our contention that Pharma preferentially make payments to senior individuals who occupy influential posts and therefore higher payments tend to be directed to males.

(Line 338 in the revised manuscript)

“Although there are 1.25 times more male dermatologists than female dermatologists (3189 male dermatologists and 2543 female dermatologist) in Japan [18], the lower proportion of female CPG authors and lower Pharma payments to female CPG authors could not be explained merely by the discrepancy in numbers alone. Dermatology is one of the most attractive medical specialties for females in Japan, as shown by the fact that, overall, only 21% of Japanese doctors are female whereas the figure in dermatology is more than double of that (44%). Dermatology offers better and more amenable job opportunities for women, such as working hours (i.e. a normal 9-5 working day with little overtime), less exhausting work, few if any invasive procedures and far less prospect of being sued for malpractice [19]. This allows women to, as far as possible, follow the society-driven role of being a homemaker, as well as maintain their employment as a physician. Nonetheless the status of females in Japan has been traditionally much lower compared with male counterparts with little recent tangible improvement [20, 21]. There are long-standing and profound prejudices regarding females in the male-dominated and patriarchal Japanese medical community. The nation discriminates against female physicians, with Japan’s total of female doctors being the lowest among industrialized nations. As a recent illustration of the degree of discrimination against females in the medical field in Japan, in 2018, it became evident that many medical schools were suppressing the number of female medical students by manipulating entrance examination scores to ensure that many women could not gain entrance to medical schools while males with lower scores were accepted [22]. The lack of gender equality in all aspects of life in Japanese society is manifest in the prevailing belief by the male-dominated hierarchy that a woman’s role is to get married and become a housewife and raise children [23-25]. Consequently, in the medical field, it is assumed that any women qualifying as doctors will reasonably quickly relinquish their posts to marry, commit to domestic duties and raise their children, resulting in a waste of resources needed to educate and train them and difficulties in replacing them when they quit. Partially supported by these prejudices against females, in Japan’s male dominated society, male physicians usually hold higher academic positions, such as directors of hospital and chairpersons of CPG committees. Therefore, Pharma may concentrate their activities on male CPG authors who are in influential positions rather than on female ones.”

Also is there any data or studies from other specialty groups in Japan by which to compare the dermatologist payments?

Reply:

Our team of authors has reported financial relationships between pharmaceutical companies and CPG authors in several specialties, such as oncology, dementia, infectious disease and orthopedics, and we have added some text in this respect: 

(Line 282 in the revised manuscript)

“In the present study, we found that dermatology CPG authors received $13,300 (SD: $20,475) in the mean monetary value of payment per individual per year. Saito et al. reported the mean monetary values of payment per author per year were $10,565 (SD: $20,059) for oncology CPG authors and $11,568 (SD: $16,874) for orthopedic surgery professors in Japan [10, 15]. Given that Executive Board members of the JDA received the second highest payments in the median values among those representing 18 basic medical fields in Japan [5], these figures suggested Japanese Pharma might focus more attention on dermatology than other medical fields.”

---

## [Decision Letter · Decision Letter 1]

10 Sep 2020

Pharmaceutical company payments to dermatology Clinical Practice Guideline authors in Japan.

PONE-D-20-12296R1

Dear Dr. Murayama,

We’re pleased to inform you that your manuscript has been judged scientifically suitable for publication and will be formally accepted for publication once it meets all outstanding technical requirements.

Kind regards,

Tim Mathes

Academic Editor

PLOS ONE

Additional Editor Comments (optional):

Reviewers' comments:

Reviewer's Responses to Questions

**Comments to the Author**

1. If the authors have adequately addressed your comments raised in a previous round of review and you feel that this manuscript is now acceptable for publication, you may indicate that here to bypass the “Comments to the Author” section, enter your conflict of interest statement in the “Confidential to Editor” section, and submit your "Accept" recommendation.

Reviewer #1: All comments have been addressed

2. Is the manuscript technically sound, and do the data support the conclusions?

Reviewer #1: Yes

3. Has the statistical analysis been performed appropriately and rigorously? 

Reviewer #1: Yes

4. Have the authors made all data underlying the findings in their manuscript fully available?

Reviewer #1: Yes

5. Is the manuscript presented in an intelligible fashion and written in standard English?

Reviewer #1: Yes

6. Review Comments to the Author

Reviewer #1: (No Response)

7. PLOS authors have the option to publish the peer review history of their article (what does this mean?). If published, this will include your full peer review and any attached files.

Reviewer #1: **Yes: **Jake Checketts, DO

---

## [Editor Report · Acceptance letter]

30 Sep 2020

PONE-D-20-12296R1 

Pharmaceutical company payments to dermatology Clinical Practice Guideline authors in Japan. 

Dear Dr. Murayama:

I'm pleased to inform you that your manuscript has been deemed suitable for publication in PLOS ONE. Congratulations! Your manuscript is now with our production department. 

Kind regards, 

on behalf of

Dr. Tim Mathes 

Academic Editor

PLOS ONE